# Generative Image Compression by Estimating Gradients of the Rate-variable Feature Distribution

## Abstract

While learned image compression (LIC) focuses on efficient data transmission, generative image compression (GIC) extends this framework by integrating generative modeling to produce photo-realistic reconstructed images. In this paper, we propose a novel diffusion-based generative modeling framework tailored for generative image compression. Unlike prior diffusion-based approaches that indirectly exploit diffusion modeling, we reinterpret the compression process itself as a forward diffusion path governed by stochastic differential equations (SDEs). A reverse neural network is trained to reconstruct images by reversing the compression process directly, without requiring Gaussian noise initialization. This approach achieves smooth rate adjustment and photo-realistic reconstructions with only a minimal number of sampling steps. Extensive experiments on benchmark datasets demonstrate that our method outperforms existing generative image compression approaches across a range of metrics, including perceptual distortion, statistical fidelity, and no-reference quality assessments.

## 1 Introduction

Image compression techniques aim to encode images into the shortest possible bit streams for efficient data transmission. Recent studies have developed learned image compression (LIC) methods (Ballé et al., 2016; 2018; Minnen et al., 2018; Minnen & Singh, 2020; Cheng et al., 2020; He et al., 2022; Liu et al., 2023; Han et al., 2024; Lu et al., 2025; Qian et al., 2022) that achieve superior rate-distortion performance compared to conventional codecs (Bellard, 2015; JVET, 2025). However, the compression process tends to sacrifice the details in images, and the optimization objective, i.e., the rate-distortion loss function, restricts the ability of decoder to restore the lost image details, which finally results in a blurred, unrealistic recovered image. The pursuit of realism has given rise to a range of generative image compression methods.

Generative image compression (GIC) methods introduce impressive generative modeling techniques, e.g., generative adversarial networks (GANs) (Goodfellow et al., 2014), vector-quantized variational autoencoders (VQ-VAEs) (Esser et al., 2021), and diffusion-based models (Song & Ermon, 2019; Ho et al., 2020), to obtain the capability of prior distribution modeling, thus improving human-perceptual performance with guaranteed fidelity. In their seminal work, perceptual loss, GANs (Agustsson et al., 2019; Mentzer et al., 2020; Tschannen et al., 2018), and their discriminator variants (Muckley et al., 2023) are first utilized to finetune the basic image compression network, which allows the decoder in the autoencoder to complement the image details. For the VQ-based method (Mao et al., 2024; Jia et al., 2024; Li et al., 2024), the prior distribution of compressed latent variables is modeled as classification probabilities according to the codebook. Compared to GAN-based and VQ-based methods, diffusion modeling decouples their once-through distribution transformations into asymptotic stochastic processes, which significantly enhances generation performance. The existing diffusion-based methods (Yang & Mandt, 2023; Hoogeboom et al., 2023; Careil et al., 2023) can be viewed as a form of "post-processing", where diffusion models are employed to enhance the compressed data—an indirect approach aimed at supplementing the lost details. However, such an indirect method may not fully harness the potential of diffusion modeling.

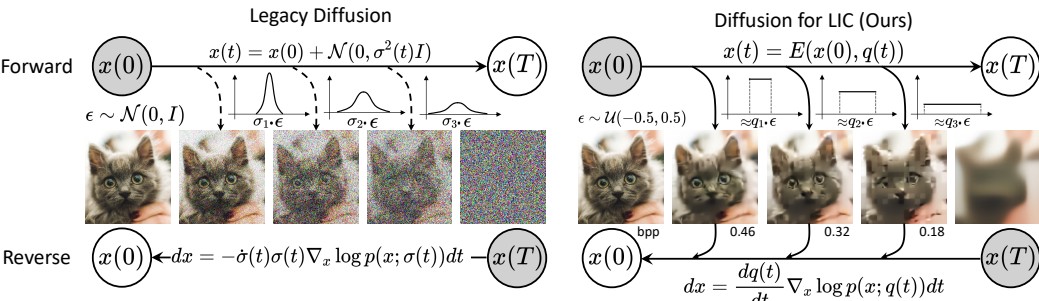

Figure 1: **Left:** The forward and reverse process of legacy diffusion for comparison. The legacy diffusion consists of transforming data to a simple noise distribution and a reverse ODE to restore the original data. **Right: The overview pipeline of our method.** The forward process is defined as the entropy model compressing the data. With the bit rates decrease, the compressed images retain less details (please *zoom in* for better visualization). We can reverse such ODE at any intermediate time to recover the data under various compression rates. This makes a full use of the benefits of diffusion modeling and an organic integration of LIC and diffusion.

Revisiting diffusion modeling in the context of image generation, a forward process is constructed by progressively corrupting data with increasing Gaussian noise, and generative modeling is achieved by training a sequence of probabilistic models to reverse the corruption process. Rate-variable quantization, i.e., to quantize features with different quantization factors, could establish a similar process of gradually distorting high-quality data. As shown in Fig. 1, legacy diffusion model adds Gaussian noise to distort the data while the corruption process of rate-variable quantization is often formulated as additive noise with uniform distribution (Ballé et al., 2016; 2018). Following this, we argue that the rate-variable compression forms a particular forward process with additive noise, and that the goal of restoring the details oriented to the state before compression can be achieved by reversing it. Taking inspiration from generative modeling through stochastic differential equations (SDEs) (Song et al., 2020b), we describe the aforementioned forward process (compression) and reverse process with the help of SDEs. That is, it is possible to construct intermediate probability distributions for a stochastic process using a variety of compression rates, and we train a neural network to model the score of a compression-rate-dependent marginal distribution of the training data corrupted by compressing. Moreover, with the adopted rate-variable quantization parameterizing the compression process into a single quantization factor, this exactly facilitates the formulation of our customized forward and reverse processes. The above framework establishes a sequence of feature distributions, resulting in a more natural integration of diffusion model and learned image compression, which supports rate-variable generative compression with only a minimal number of reverse steps.

Taken together, we propose generative image compression by estimating gradients of rate-variable feature distribution, offering a novel perspective beyond conventional paradigms. Our method constitutes a customized diffusion framework designed for image compression, featuring both **flexible rate adjustment** and **high-fidelity image reconstruction**. Our rate-variable model is able to outperform current fixed-rate state-of-the-art methods on a range of perceptual metrics. We believe that this work will inspire further innovations in broader areas, particularly in the modification of the diffusion modeling core to suit a variety of specific research scenarios.

## 2 BACKGROUND

In this section, we present the related work and the existing technologies that we adopt in our proposed method. First, we review a brief theory of denoising score matching modeling (Song & Ermon, 2019) as a preliminary for our specialized diffusion models. Next, we describe the adopted rate-variable compression strategy for the sake of understanding our proposed framework. Finally, we introduce recent representative GIC methods that integrate with generative modeling.

### 2.1 DENOISING SCORE MATCHING

Given the data distribution $p_{data}(\boldsymbol{x})$, the idea of score matching is to find the score/gradients of the data distribution $p_{data}(\boldsymbol{x})$, that is, the fastest growing direction of the log probability density of

the data $s(\boldsymbol{x}) = \nabla_{\boldsymbol{x}} \log p(\boldsymbol{x})$. To obtain the score, Song & Ermon (2019) proposed the denoising score matching and noise conditional score networks (NCSN): To consider the family of mollified distributions $p(\boldsymbol{x}; \sigma)$ obtained by adding i.i.d. Gaussian noise of standard deviation $\sigma$ to the data, the diffusion model sequentially denoises from a pure Gaussian noise $\boldsymbol{x}_0 \sim \mathcal{N}(\mathbf{0}, \sigma_0^2 \mathbf{I})$ into intermediate states $\boldsymbol{x}_i \sim p(\boldsymbol{x}; \sigma_i)$ with decreasing noise levels $\sigma_i > \sigma_{i+1}$. It ends up with the target image $\boldsymbol{x}_N$ with $\sigma_N = 0$ at the place where the log probability density is maximized. The optimization objective is to minimize the $L_2$ denoising error for samples drawn from $p_{data}$ separately for every $\sigma$. Defining a neural denoiser $D(\boldsymbol{x}, \sigma)$:

$$\mathbb{E}_{\boldsymbol{y} \sim p_{data}} \mathbb{E}_{\boldsymbol{n} \sim \mathcal{N}(\mathbf{0}, \sigma^2 \mathbf{I})} ||D(\boldsymbol{y} + \boldsymbol{n}, \sigma) - \boldsymbol{y}||_2^2, \tag{1}$$

where $\boldsymbol{y}$ is a training image and $\boldsymbol{n}$ is the sampled Gaussian noise. In this vein, the score of the state with noise of deviation $\sigma$ can be formulated as:

$$\nabla_{\boldsymbol{x}} \log p(\boldsymbol{x}; \sigma) = (D(\boldsymbol{x}; \sigma) - \boldsymbol{x})/\sigma^2. \tag{2}$$

For the sampling of the diffusion modeling above, Song et al. (2020b) present a stochastic differential equation (SDE) to unify the processes of noise removal and addition into an integral theoretical framework. An overview of legacy diffusion with a simplified form of SDEs, ordinary differential equations (ODEs), is shown in Fig. 1. To solve the ODE is to substitute Eq. 2 into it and calculate the numerical integration, i.e., taking finite steps over discrete time intervals.

## 2.2 Learned Lossy Image Compression

Learned lossy image compression methods are built upon a variational autoencoder (VAE) framework proposed by Ballé et al. (2018). The VAE based LIC framework mainly comprises an autoencoder and an entropy model. The autoencoder conducts nonlinear transforms between the image space, i.e., input: $\boldsymbol{x}$, reconstruction: $\hat{\boldsymbol{x}}$, and the latent representation space, i.e., latent representation: $\boldsymbol{y}$, quantized latents: $\hat{\boldsymbol{y}}$; while, the entropy model minimizes the code length by estimating the probability distribution of latent representations.

**Rate-distortion optimization.** In their seminal work, Ballé et al. (2016) established the end-to-end rate-distortion minimization framework. It showed that the smallest average code length of latent representation is given by the Shannon cross entropy (Shannon, 1948) between the actual marginal distribution and a learned entropy model. Thus, the optimization objective appears a rate-distortion trade-off between rate loss $R(\cdot)$ and distortion loss $D(\cdot)$:

$$\mathcal{L}_{R-D} = \mathcal{R}(\hat{\boldsymbol{y}}) + \lambda \cdot \mathcal{D}(\hat{\boldsymbol{x}}, \boldsymbol{x}), \tag{3}$$

where $\lambda$ controls whether the network is more concerned about the quality of the recovery or the compression efficiency. The optimization problem under the fixed $\lambda$ in Eq. 3 is employed for the fixed-rate paradigm, driving the encoder to adjust the information reserved in the latent variables $\boldsymbol{y}$.

**Quantization scaling.** Nevertheless, the LIC network trained under this schedule only yields a single compression rate result, which limits the design space of diffusion modeling. A general solution is to randomly sample $\lambda$ during training, finetuning a vanilla fixed-rate LIC network to support rate-variable compression (Toderici et al., 2015; Choi et al., 2019; Cui et al., 2021; Wang et al., 2023a). In this work, we adopt quantization scaling to control the compression rate via the entropy model. Under this circumstance, the role of "information reducer" moves from the encoder to the entropy model, separating the autoencoder as a stand-alone component. This strategy focuses on the quantization operation $\lceil \boldsymbol{y} \rfloor$. Since codecs only work on integers, the entropy model quantized the latent representation for bit stream transmission. The quantization operation can be regarded as adding a uniform noise in a range of $[-0.5, 0.5]$:

$$\hat{\boldsymbol{y}} = \lceil \boldsymbol{y} \rfloor \Rightarrow \boldsymbol{y} + \mathcal{U}(-0.5, 0.5). \tag{4}$$

The idea of quantization scaling is to scale the latent representation $\boldsymbol{y}$ before quantization. Given a scale parameter $q$, this can be formulated as:

$$\begin{aligned}
\hat{\boldsymbol{y}}_q &= \lceil \boldsymbol{y}/q \rfloor \cdot q \\
&\Rightarrow [(\boldsymbol{y}/q) + \mathcal{U}(-0.5, 0.5)] \cdot q \\
&= \boldsymbol{y} + \mathcal{U}(-0.5, 0.5) \cdot q.
\end{aligned} \tag{5}$$

Referring to Eq. 4, quantization scaling can be considered as scaling the uniform noise to control the information gap between original latents $\boldsymbol{y}$ and quantized latents $\hat{\boldsymbol{y}}_q$. At this point, we obtain a rate-variable compression network with only one parameter to adjust the compression rate. This facilitates our following theory construction and its implementation.

**Generative image compression.** Generative modeling methods, e.g., generative adversarial nets (GANs), vector-quantized variational autoencoders (VQ-VAEs), and diffusion-based models, probabilistically model real data distributions. Generative compression methods exploit them to supplement the prior distribution to the compressed data, thereby producing photo-realistic reconstructed images. In their seminal work, GANs and perceptual loss are first utilized to enhance the fidelity of the reconstructed image. Agustsson et al. (2019) first introduced GANs to generate extra details for shaper decompressed images. Subsequent work has studied further the variants of discriminators, such as patch-GAN (Mentzer et al., 2020), local label prediction (Muckley et al., 2023), and realism guidance (Agustsson et al., 2023). In the context of the VQ-based method, a vector-quantized variational autoencoder is used to replace or wrap the vanilla LIC variational autoencoder. Jia et al. (2024) move the lossy compression framework into the latent space of VQ-VAEs. Li et al. (2024) modify the basic VQ-VAEs to obtain a rate-variable generative image compression network. Given the surprising results of diffusion modeling in the generative domain, recent work has attempted to incorporate the advantages of the diffusion model into image compression. The existing diffusion-based methods employ two primary training approaches. The first approach involves training a denoising network itself (Yang & Mandt, 2023; Hoogeboom et al., 2023), while the second approach involves fine-tuning a pretrained large-scale diffusion model (Careil et al., 2023; Lei et al., 2023). With respect to the integration of diffusion models within the LIC network, the prevailing methodologies encompass two approaches: the replacement of the decoder in the compression autoencoder with the diffusion model (Yang & Mandt, 2023), and the subsequent addition of the denoising network following the completion of the compression procedure (Hoogeboom et al., 2023). Existing diffusion-based methods indirectly take advantage of diffusion modeling. We expect to establish a novel framework to seamlessly combine LIC and diffusion, realizing the potential of diffusion modeling. Deriving from the nature of the diffusion modeling is a delicate way.

## 3 METHOD

In the context of learned image compression technologies, the loss of detailed information occurs in the encoder; while the decoder decodes the broken data to images. Restricted to the training strategy and network paradigm, it is difficult for the decoder to restore the lost image details. We propose generative image compression by estimating gradients of rate-variable feature distribution to reverse the compression process, assisting the restoration of lost details. Note that we consider our proposed diffusion modeling specialized in image compression to be regarded as a form of "generalized diffusion". Consequently, our analysis lives outside the confines of the diffusion theoretical frameworks but borrows some analytical processes and ideas from these frameworks (Song & Ermon, 2019; Song et al., 2020b; Karras et al., 2022). Building a standard diffusion model involves two key components, i.e., degraded data construction for training a denoising network (forward process) and the sampling design (reverse process). In this section, we construct the diffusion modeling specialized in image compression following the above modules.

### 3.1 COMPRESSION FORWARD PROCESS

Given an image $\boldsymbol{x}_0$, the legacy diffusion obtains degraded data $\boldsymbol{x}_i$ by adding various levels (denoted by deviations $\sigma_N = \sigma_{max} > \cdots > \sigma_1 > \sigma_0 = 0$) of Gaussian noise so that $p(\boldsymbol{x}_i|\boldsymbol{x}_0) \sim \mathcal{N}(\boldsymbol{x}_0, \sigma_i^2 \mathbf{I})$. This corrupts data to varying degrees, and the network is trained to estimate the score function $\nabla_{\boldsymbol{x}} \log p(\boldsymbol{x}_i)$ to restore the original data. In essence, learning to restore from corrupted data, also known as the reverse process, enables the network to perform score matching. The training of the reverse process is relaxed and facilitated by data at various levels of corruption.

For the image compression task, we expect to equip the reverse neural network with the ability to restore the compression-corrupted data. Following the standard diffusion process, we replace the data corruption of adding noises with rate-variable compression. Defining a pretrained rate-variable

entropy model $E$, the compression process can be formulated as:

$$\boldsymbol{x}_t = E(\boldsymbol{x}_0, q_t), \tag{6}$$

where $q_t$ denotes the parameter of quantization scaling as we mentioned in Sec 2.2. Similar to the deviation $\sigma$ in legacy diffusion, $q$ reflects the extent of data corruption.

The reverse neural network $D_\theta$ inverts the corruption due to compression:

$$\hat{\boldsymbol{x}}_0 = D_\theta(\boldsymbol{x}_t, q_t), \tag{7}$$

where $\hat{\boldsymbol{x}}_0$ denotes the approximated recovered data produced by network $D_\theta$. When $q_t$ is small, $\hat{\boldsymbol{x}}_0$ should appear close to real $\boldsymbol{x}_0$ and vice versa. Thus, our optimization objective is to minimize the distance between ground truth $\boldsymbol{x}_0$ and approximated $\hat{\boldsymbol{x}}_0$:

$$\mathbb{E}_{\boldsymbol{x}_0 \sim p_{data}, q_t \sim \epsilon_q} ||\boldsymbol{x}_0 - D_\theta(E(\boldsymbol{x}_0, q_t), q_t)||^2, \tag{8}$$

where $\boldsymbol{x}_0$ is randomly drawn from a dataset and $q_t$ is sampled from the distribution $\epsilon_q$. Due to the equal significance of all bit rates in the context of rate-variable compression tasks, we set the distribution $\epsilon_q := \mathcal{U}(q_{min}, q_{max})$, where $q_{max}$ is the maximum supported scale parameter for the entropy model and $q_{min}$ is lower than the minimum support, a constant very close to $0$. When the sampled $q_t$ is not supported by the entropy model, referring to Sec. 2.2 we take a simulated quantization method as $\boldsymbol{x}_t = \boldsymbol{x}_0 + \mathcal{U}(-0.5, 0.5) \cdot q_t$.

## 3.2 REVERSE PROCESS DESIGN

In their pioneering work, Song et al. (2020b) present a stochastic differential equation (SDE) that maintains the desired distribution $p$ as the sample $\boldsymbol{x}$ evolves over time. Within this theoretical framework, the sampling process is defined as a reverse SDE, which facilitates the derivation of sampling formulas oriented to better generation results. Following this, we endeavor to formulate a sampling approach that aligns with the diffusion modeling of our design for image compression. To that end, we first express the aforementioned diffusion process as an ordinary differential equation (ODE). Subsequently, a reverse ODE is derived and extended to SDEs for enhancing the quality of the reconstructed images.

**ODE formulation.** Although our diffusion forward process is implemented by the entropy model $E$, we can approximate it (see Sec. 2.2) as follows:

$$\boldsymbol{x}_t \Rightarrow \lceil \boldsymbol{x}_0 / q(t) \rfloor \cdot q(t) \Rightarrow \boldsymbol{x}_0 + \mathcal{U}(-0.5, 0.5) \cdot q(t). \tag{9}$$

For the sake of representing differential equations, we move the subscript $q_t$ into the function brackets $q(t)$. We define the ODE evolving a sample $\boldsymbol{x}_a \sim p(\boldsymbol{x}_a; q(t_a))$ from time $t_a$ to $t_b$ yields a sample $\boldsymbol{x}_b \sim p(\boldsymbol{x}_b; q(t_b))$, which is satisfied by

$$d\boldsymbol{x} = -\frac{dq(t)}{dt} \nabla_{\boldsymbol{x}} \log p(\boldsymbol{x}; q(t)) dt, \tag{10}$$

where $\nabla_{\boldsymbol{x}} \log p(\boldsymbol{x}; q(t))$ is the score function in our theoretical framework. To reverse it, we define the reverse neural network $D_\theta$ producing a result $\hat{\boldsymbol{x}}_0$ approximated to $\boldsymbol{x}_0$. Thus, the score is obtained by $\nabla_{\boldsymbol{x}} \log p(\boldsymbol{x}; q(t)) = (\hat{\boldsymbol{x}}_0 - \boldsymbol{x}_t)/q(t)$. Following the previous works, the Euler's method is adopted as the discrete solution during sampling. We substitute the equation above to Eq. 10 and use Euler's solver (see Appendix B):

$$\boldsymbol{x}_{i+1} = \boldsymbol{x}_i + \frac{q(t_i) - q(t_{i+1})}{q(t_i)} (\hat{\boldsymbol{x}}_0 - \boldsymbol{x}_i). \tag{11}$$

Nevertheless, the reverse ODE in isolation remains not an optimal solution for sampling, since we find that the deterministic sampling based on the derivation of ODE produces a suboptimal image quality compared to SDE-based stochastic sampling.

**Stochastic sampling.** The reverse ODE, corresponding to deterministic sampling, has been observed to result in a worse performance (Song et al., 2020a;b) than stochastic sampling, i.e., reverse SDEs. Following the existing work (Karras et al., 2022), we extend the ODEs Eq. 10 to SDEs and append the random items from Langevin sampling:

$$d\boldsymbol{x} = -\frac{dq(t)}{dt} \nabla_{\boldsymbol{x}} \log p(\boldsymbol{x}; q(t)) dt + \alpha(t) d\omega_t \pm \alpha(t) \nabla_{\boldsymbol{x}} \log p(\boldsymbol{x}; q(t)), \tag{12}$$

---

**Algorithm 1:** The procedure of our diffusion-based compression of data $x$

---

**Input:** $x, q_0$   **Given:** $\mathcal{E}(x), \mathcal{D}(\bar{y}), E_\phi(y, q), D_\theta(\hat{y}, q), q_{i\in\{0,\cdots,N\}}, \alpha, \omega$

1 **Encoding**                                                                    $\triangleright$ $x$ to be compressed, $q_0$ setting the compression rate
2     $y_0 \leftarrow \mathcal{E}(x)$
3     $\hat{y}_q \leftarrow E_\phi(y_0, q_0)$                                    $\triangleright$ Approximate $p(\hat{y}_q)$ and compress $y_0$ with scale $q_0$
4   $\hat{y}_q \xleftarrow{p(\hat{y}_q)}$ bit stream                            $\triangleright$ Entropy code using $p(\hat{y}_q)$
5 **Decoding**                                                                      $\triangleright$ Reverse directly from $\bar{y}_0 := \hat{y}_q$
6     **for** $i \in \{0, \cdots, N-1\}$ **do**
7         $d_i \leftarrow (D_\theta(\bar{y}_i, q_i) - \bar{y}_i)/q_i$                        $\triangleright$ Evaluate the score $\nabla_x \log p(x; q_i)$ at $q_i$
8         $\bar{y}_{i+1} \leftarrow \bar{y}_i + (q_i - q_{i+1})d_i$                        $\triangleright$ Take Euler step from $q_i$ to $q_{i+1}$
9         **sample** $\epsilon_i \sim \omega$
10       $\bar{y}_{i+1} \leftarrow \alpha(\epsilon_i - d_i)$                        $\triangleright$ Inject randomness for stochastic sampling
11     **end**
12    $\hat{x} \leftarrow \mathcal{D}(\bar{y}_N)$
13 **return** $\hat{x}$

---

where $\omega_t$ is the standard Wiener process to inject the randomness to the sampling, the last term is the deterministic decaying item, and $\alpha(t)$ is the hyperparameter set empirically. Generally speaking, randomness injection is implemented by adding standard Gaussian noises in the legacy diffusion model framework (Song & Ermon, 2019); however, our framework lives outside the standard diffusion theory, so we discuss the form of randomness injection and hyperparameter schedules in the following section.

**Randomness injection.**   Previous results (Song et al., 2020a;b) show that finding an optimal setting of stochasticity is significant and that the setting should be empirically determined with respect to specific diffusion models. In legacy diffusion models, the amount of stochasticity grows with the number of sampling steps. Recent work (Karras et al., 2022) suggests that a non-uniform growing schedule surpasses a linear one. Combining the above with the properties of compression tasks, we propose a randomness injection schedule for the hyperparameter $\alpha_t$:

$$\alpha_t = \beta \cdot \sqrt{q_t - q_{min}}, \tag{13}$$

where $q_t$ adjusts the compression rate, $q_{min}$ is the minimus quantization scale parameter supported by the entropy model, and $\beta$ controls the growth rate of stochastic injections with $q_t$. Moreover, due to the specialization of our model, the specific form of stochasticity is worth arguing, e.g., standard Gaussian noise, uniform noise, and noise drawn from the entropy model. Our analysis in Sec. 4.2 discuss the setting of $\beta$ and the form of randomness injection $\omega$.

### 3.3 ESTIMATING GRADIENTS OF RATE-VARIABLE FEATURE DISTRIBUTION

**Training the rate-variable compression entropy model.**   To alleviate the computational overhead, we move the compression entropy model and the diffusion network into the latent space of the pretrained VAE (Rombach et al., 2022), where the original image is compressed initially by a factor of four in spatial dimensions. We train the rate-variable entropy model $E_\phi$ via optimizing a rate-distortion loss:

$$\mathcal{L}_{R-D} = -\log p(\hat{z}) - \log p(\hat{y}) + \lambda \cdot ||\hat{y} - y||^2, \tag{14}$$

where $y$ is the latent representation produced by VAE encoder $\mathcal{E}(x)$, $\hat{z}$ is the hyperprior latents, and $\hat{y}$ is quantized by $E_\phi(y, q)$. For multi-rate training, we randomly sample $\lambda$ and obtain the corresponding $q$. The network architecture of the entropy model is established in accordance with (Han et al., 2024), the most recent SOTA work that attains a great trade-off between inference latency and rate-distortion performance. Further elaborations about such compression paradigms can be found in the literatures (Ballé et al., 2016; 2018; Han et al., 2024).

**Training the reverse neural network.**   Inspired by the recent work (Karras et al., 2022), we borrow the denoising U-net architecture from it. Following Sec. 3.1, in every training iteration a quantization scale $q_t$ is sampled from the uniform distribution $q_t \sim \mathcal{U}(q_{min}, q_{max})$. Given a latent

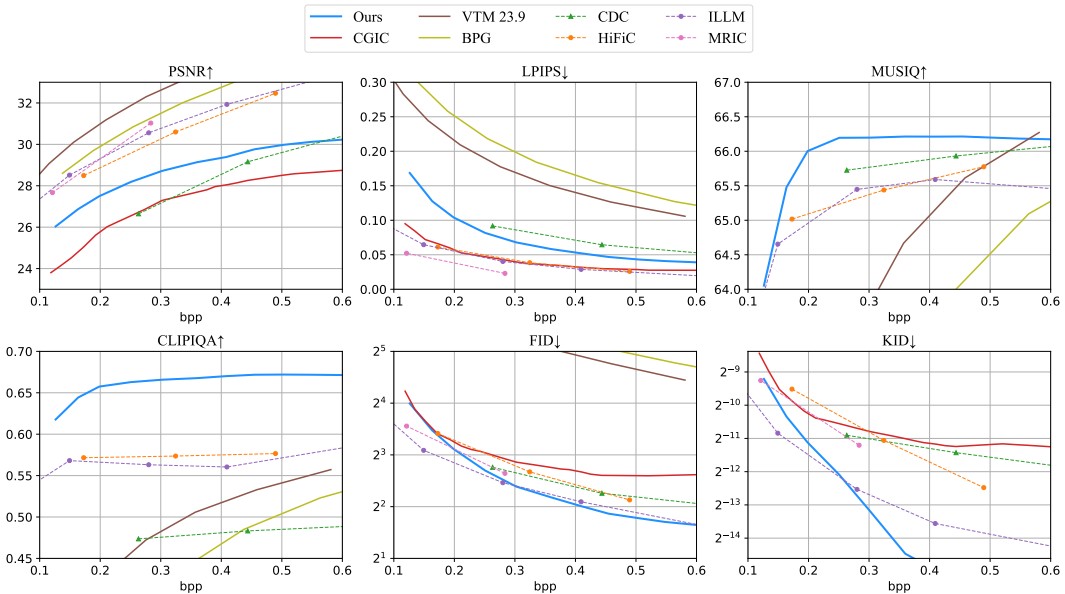

Figure 2: Comparisons of methods across various distortion and statistical fidelity metrics for the DIV2K dataset. The continuous lines represent rate-variable methods (one model for all bit rates). Circular markers denote GAN-based methods and triangular markers denote diffusion-based methods (every marker corresponds to a separate model respectively).

representation $\boldsymbol{y}_0$ extracted from a sampled image $\boldsymbol{x}$, the compressed latent representation $\boldsymbol{y}_t$ is obtained by the entropy model $\boldsymbol{y}_t = E_\phi(\boldsymbol{y}_0, q_t)$. The reverse neural network $D_\theta$ is trained using a single $L_2$ distance:

$$\mathcal{L}_{\text{diff}} = ||\boldsymbol{y}_0 - D_\theta(\boldsymbol{y}_t, q_t)||^2. \tag{15}$$

In order to maintain the purity of the theoretical framework, this work incorporates no generative adversarial networks (GANs) or perceptual loss finetuning stage. Nonetheless, the capacity to generate photo-realistic images of the proposed approach is evidenced by its exceptional performance on a range of perceptual metrics (see Experiment Sec. 4.3).

**Putting it together.**  The whole process of the proposed Algorithm 1 can be elaborated as: Given a source image vector $\boldsymbol{x}$, the autoencoder contains a parametric analysis transform $\mathcal{E}$ to obtain the latent representation $\boldsymbol{y}_0$ from $\boldsymbol{x}$ and a parametric synthesis transform $\mathcal{D}$ for reconstruction. $\boldsymbol{y}$ is then quantized and compressed by the entropy model $E_\phi$ with a quantization scale of $q_0$ to latents $\hat{\boldsymbol{y}}_q$ for storage or transmission. When decoding to the reconstructed image $\hat{\boldsymbol{x}}$, a reverse network $D_\theta$ is utilized to reverse directly from the compressed data $\hat{\boldsymbol{y}}_q$ for minimal steps. The reversed latent variable $\bar{\boldsymbol{y}}_N$ is then fed to the parametric synthesis transform $\mathcal{D}$ for reconstruction $\hat{\boldsymbol{x}}$.

## 4 Experiments

### 4.1 Experimental Settings

**Datasets.**  For the first training stage, we follow the previous compression work (Han et al., 2024) and train the entropy model on the Open Images (Kuznetsova et al., 2020) dataset. The randomly selected Open Images dataset contains 300k images with a short edge of no less than 256 pixels. For the second training stage, we follow the previous diffusion work (Karras et al., 2022) and train the reverse neural network on the training set of ImageNet (Deng et al., 2009). For evaluation, three benchmarks, i.e., DIV2K dataset (Agustsson & Timofte, 2017), Kodak image set (Kodak, 1993), and CLIC2020 test set (Toderici et al., 2020), are utilized to evaluate the proposed network.

**Implementation details.** The detailed architecture and hyperparameter settings of the proposed entropy model refer to the previous work (Han et al., 2024) and Appendix F. For the reverse neural network, we employ the U-Net structure from EDM (Karras et al., 2022). Our experiments and evaluations are carried out on Intel Xeon Platinum 8375C and Nvidia RTX 4090 graphics cards. By default, our proposed networks are trained using the AdamW optimizer (Loshchilov & Hutter, 2017). The weight decay and momenta for AdamW are 0.02 and (0.9, 0.95). For the entropy model, we randomly crop $256 \times 256$ sub-blocks from the Open Images dataset (Kuznetsova et al., 2020) with a batch size of 8. We train the entropy model in two stages. In the first (fixed-rate) stage, the model is trained for 0.75M steps using a constant learning rate of $1e-4$. In the second (multi-rate) stage, training continues for another 0.75M steps and then decreases the learning rate to $1e-5$ for 0.375M steps. The network is optimized with the MSE metric, which represents the last term in Eq. 14. For multi-rate training, the multipliers $\lambda$ are $\{0.05, 0.025, 0.01, 0.005, 0.001, 0.0005, 0.0001\}$. As for the reverse neural network, we crop $256 \times 256$ center blocks from the ImageNet dataset (Deng et al., 2009) with a batch size of 128. We optimize the network with the initial learning rate $1e-4$ for 0.5M steps and then decrease the learning rate to $5e-5$ for the remaining 0.5M steps. Since the compression task provides a strong prior (the compressed data), minimal sampling steps are required for the reverse network during the decoding process. We only use 2 reverse steps for all bit rates, which improves the efficiency of our method.

**Comparison methods and metrics.** We compare our method with the hand-crafted coding standards VVC (JVET, 2025), BPG (Bellard, 2015) and recent state-of-the-art methods (Muckley et al., 2023; Mentzer et al., 2020; Yang & Mandt, 2023; Li et al., 2024; Agustsson et al., 2023). Note that our method should be classified with CGIC (Li et al., 2024), a category of **rate-variable generative image compression**. The rate-variable methods obtain compressed results at all bit rates using only one model, while the other compared methods are fixed-rate, i.e., multiple separate models are required to be retrained for various compression rates. CDC (Yang & Mandt, 2023) is the most recent state-of-the-art **diffusion-based method**. Other methods: HiFiC (Mentzer et al., 2020), MRIC (Agustsson et al., 2023) and ILLM (Muckley et al., 2023) are **GAN-based approaches**. We use bits per pixel (bpp) value to indicate the compression ratio. In addition to the basic distortion metric PSNR, a range of perceptual metrics are used as evaluation: perceptual distortion: LPIPS (Zhang et al., 2018), non-reference measure: MUSIQ (Ke et al., 2021), CLIPIQA (Wang et al., 2023b), and statistical fidelity: FID (Heusel et al., 2017), KID (Bińkowski et al., 2018). For the calculation of FID and KID, we follow the previous work (Mentzer et al., 2020) to patchify the high-resolution images into subimages of size $256 \times 256$.

### 4.2 MODEL ANALYSIS

**Form of randomness $\omega$.** We consider three forms of randomness: Gaussian noise, which is the general choice in legacy diffusion models; uniform noise, which simulates the quantization process; and noise sampled from the probability distribution of latent variables estimated by the entropy model. Fig. 4 shows the impacts of various types of randomness. We find that Gaussian noise and uniform noise are comparable, while noise from the entropy model performs worse. Consequently, we adopt uniform and Gaussian forms of stochasticity as our final randomness injection schedule.

**Hyperparameter $\beta$.** We further study the amount of stochasticity. Following Sec. 3.2, $\beta$ controls the growth rate of the amount of stochastic injections with $q_t$. Fig. 4 demonstrates that there exists a trade-off among perceptual distortion (LPIPS), statistical fidelity (FID), and non-reference metric (CLIPIQA). The results of Gaussian and uniform curves are analogous, and we ultimately select $\beta = 0.075$ as the final setting. More analysis on hyperparameters can be found in Appendix A.

### 4.3 COMPARISON WITH STATE-OF-THE-ART METHODS

**Rate-distortion comparison.** We evaluate the rate-distortion performance of our proposed models by drawing the rate-distortion curves Fig. 2. As DIV2K (Agustsson & Timofte, 2017) is one of the most commonly used benchmark datasets in the field of low-level vision, we mainly compare our proposed network with the aforementioned SOTA methods on DIV2K dataset. Reference models (VTM 23.9 and BPG) achieve the best PSNR scores, but display poor perceptual distortion and statistical fidelity. GAN-based methods tend to obtain a better LPIPS metric, while, compared

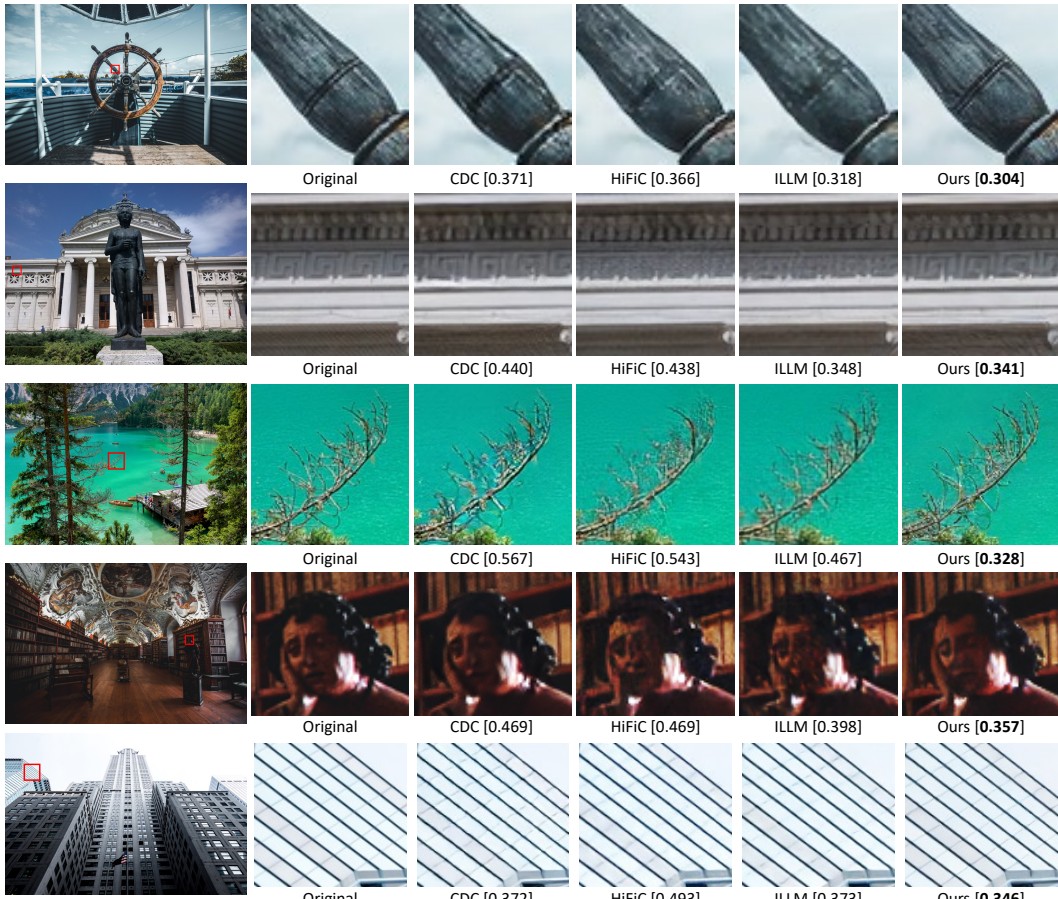

Figure 3: Visualization of the reconstructed images (top to bottom: *0824*, *0812*, *0807*, *0841*, and *0846*) from DIV2K dataset. The titles under the sub-figures are represented as "method [bpp]".

to the diffusion-based SOTA method (CDC), ours achieves superior performance. For the non-reference metrics (MUSIQ and CLIPIQA), our method is able to acquire a clear advantage over the other models. In the context of rate-variable image compression, our method has uniformly better statistical fidelity than CGIC as measured by FID and KID. Except for the lower bpp range (below 0.25), our method performs comparably to the fixed-rate SOTA model ILLM on FID metric, and demonstrates better statistical fidelity evaluated by KID. We also investigate the effectiveness of our method on the Kodak and CLIC2020 datasets, where the results are provided in our Appendices.

**Visualization analysis.** Thanks to our well-designed framework, our method achieves superior restoration of fine image details. Fig. 3 presents five comparison sets against recent state-of-the-art reconstruction models (Yang & Mandt, 2023; Mentzer et al., 2020; Muckley et al., 2023), with results generated at comparable bit rates on the DIV2K dataset. These visualizations demonstrate that our method faithfully reconstructs image details aligned with the original content, rather than introducing artificial or irrelevant textures. More high-res visualizations can be found in Appendix D.

## 5 CONCLUSION

In this work, we propose a novel diffusion modeling framework for generative image compression. We establish an organic integration of learned image compression and diffusion, building a complete diffusion framework from the forward process to the reverse process with the help of SDE theory. Our proposed method takes full advantage of the capacity of diffusion modeling, thus achieving state-of-the-art performance on a range of perceptual metrics. Furthermore, we believe this work will spark further innovations across a wide range of domains, especially by encouraging adaptations of the core diffusion modeling framework to adapt to diverse research needs.

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

# A  MODEL ANALYSIS

Table 1: Ablation study on FID of the impacts of various intermediate steps

| $q_0$ \ $q_1$ | 0.10 | 0.11 | 0.12 | 0.13 | 0.14 | 0.15 | 0.16 | 0.17 |
|---|---|---|---|---|---|---|---|---|
| 1.20 | 9.1077 | 8.7461 | 8.6560 | **8.6548** | 8.7309 | 8.7588 | 8.7612 | 8.7976 |
| 0.90 | 6.7858 | 6.6079 | 6.5039 | **6.4759** | 6.5175 | 6.5420 | 6.5389 | 6.6190 |
| 0.70 | 5.6103 | 5.3801 | **5.2867** | 5.2914 | 5.2976 | 5.2772 | 5.2936 | 5.2851 |
| 0.45 | 4.2231 | 4.0386 | 4.0412 | **4.0366** | 4.0596 | 4.0765 | 4.1009 | 4.0976 |

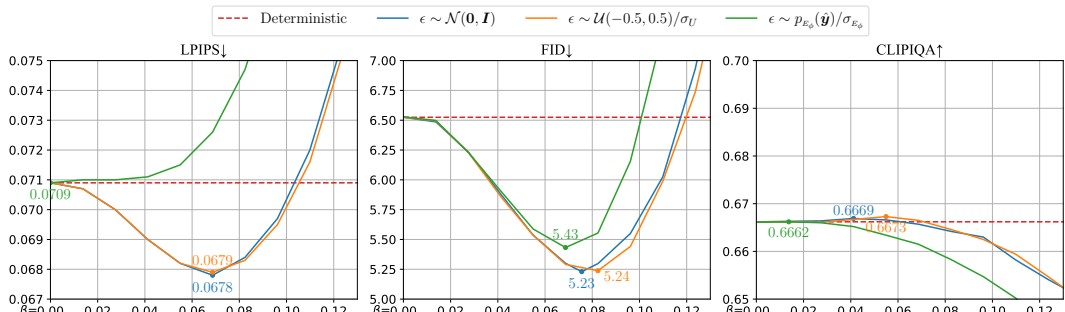

Figure 4: Evaluation of randomness injection schedules when the scale parameter $q_0$ is set as $0.7$ (i.e., bpp $= 0.3024$), test on DIV2K with LPIPS, FID, and CLIPIQA. The dashed red lines correspond to deterministic sampling, equivalent to setting $\beta = 0$. The blue, orange, and green curves correspond to drawing a noise from a standard normal distribution $\mathcal{N}$, a uniform distribution $\mathcal{U}$, and a probability distribution $p_{E_\phi}$ estimated by the entropy model $E_\phi$, respectively. Note that the latter two distributions are normalized by dividing the statistical deviations $\sigma$. The dots indicate the best observed results.

For the selection of the intermediate step, we empirically selected a fixed value of $q_1 = 0.13$ ($q_2 = 0$) because the compressed decoding part and most of the restoration tasks are essentially the last few steps of diffusion. Therefore, the intermediate value of the two-step decoding should also be set close to the noise-free (in this paper, unquantized) state to ensure that the reverse process restores more details and higher clarity. Moreover, we conduct an ablation study on FID (experiments are conducted by the full model, on the DIV2K dataset). The table 1 demonstrates that the setting of $q_1 = 0.13$ achieves a great trade-off among the various inputs $q_0$.

# B  DERIVATION OF DISCRETE ODE SOLVER

A discrete ODE solver is to use numerical methods to compute integration of ordinary differential equations. In our framework, given $\boldsymbol{x}_i$ at a compression ratio of $q_i$, we aim to obtain $\boldsymbol{x}_{i+1}$ at $q_{i+1}$. Using the first-order Euler's solver is to exploit a differential approximation:

$$\boldsymbol{x}_{t+\Delta t} = \boldsymbol{x}_t + \Delta t \cdot \frac{\mathrm{d}\boldsymbol{x}}{\mathrm{d}t}. \tag{16}$$

Substitute Eq. 10 to Eq. 16:

$$\boldsymbol{x}_{t+\Delta t} = \boldsymbol{x}_t - \Delta t \cdot \frac{\mathrm{d}q(t)}{\mathrm{d}t} \nabla_{\boldsymbol{x}} \log p(\boldsymbol{x}; q(t)). \tag{17}$$

We obtain $\nabla_{\boldsymbol{x}} \log p(\boldsymbol{x}; q(t))$ through the neural network $D_\theta$:

$$\boldsymbol{x}_{t+\Delta t} = \boldsymbol{x}_t - \Delta t \cdot \frac{\mathrm{d}q(t)}{\mathrm{d}t} \cdot \frac{\hat{\boldsymbol{x}}_0 - \boldsymbol{x}_t}{q(t)}, \tag{18}$$

$$\text{with } \hat{\boldsymbol{x}}_0 = D_\theta(\boldsymbol{x}_t, q(t)).$$

To obtain the same form as the main text, we define $t_{i+1} = t_i + \Delta t$:

$$\boldsymbol{x}_{i+1} = \boldsymbol{x}_i - (t_{i+1} - t_i) \cdot \frac{\mathrm{d}q(t)}{\mathrm{d}t} \cdot \frac{\hat{\boldsymbol{x}}_0 - \boldsymbol{x}_i}{q(t_i)}. \tag{19}$$

For simplicity and continuous sampling during training, we set $q(t) := t$:

$$\boldsymbol{x}_{i+1} = \boldsymbol{x}_i + \frac{q(t_i) - q(t_{i+1})}{q(t_i)}(\hat{\boldsymbol{x}}_0 - \boldsymbol{x}_i). \tag{20}$$

## C  COMPRESSION LATENCY

Table 2: Comparison of coding latency evaluated on Kodak dataset. All the models are evaluated on the same platform. The second line of **Model** describes the categories of the compared methods.

| Model | Ours | CDC | ILLM | CGIC |
|---|---|---|---|---|
| | Diffusion-based | Diffusion-based | GAN-based | VQ-based |
| **Enc. Time (ms)** | 123 | 23 | 60 | 85 |
| **Dec. Time (ms)** | 280 | 824 | 71 | 32 |
| **Tot. Time (ms)** | 403 | 847 | 131 | 117 |

We compare the coding efficiency of our methods with recent state-of-the-art methods (Yang & Mandt, 2023; Muckley et al., 2023; Li et al., 2024). These methods are classified into diffusion-based, GAN-based, and VQ-based approaches. As Table 2 shows, thanks to the minimal sampling steps required for our method, we achieve coding efficiency superior to that of the most recent diffusion-based SOTA work CDC. However, diffusion models generally exhibit slower coding speeds than GAN-based and VQ-based methods. The reason for this is that GAN-based methods only require a one-through transformation, whereas VQ-based methods abandon the entropy model to estimate the probability distribution, which is replaced by transmitting the index of codebook. How to further promote the inference latency of diffusion-based learned image compression methods is worth exploring in the future.

Table 3: Comparison of diffusion-based methods on coding latency evaluated on Kodak dataset. All the models are evaluated on the same platform.

| Model | CDC-17 | PerCo-5 | PerCo-20 | DiffEIC-20 | DiffEIC-50 | Ours-2 |
|---|---|---|---|---|---|---|
| **Enc. Time (ms)** | 23 | 80 | 80 | 128 | 128 | 123 |
| **Dec. Time (ms)** | 824 | 665 | 2551 | 1964 | 4574 | 280 |
| **Tot. Time (ms)** | 847 | 745 | 2631 | 2092 | 4702 | 403 |

We further report comparisons in terms of computational efficiency compared to more diffusion-based methods as [method-#steps]. It should be noted that we sample fixed 2 steps at all bit-rates so that the decoding time remains consistent with different levels of quantization. The computational complexity of our proposed method is #params: 173.457M, 201.479 GFLOPs. The compared diffusion-based GIC approaches (except for CDC) typically call large visual models. Therefore, our model is very advantageous in terms of parameter quantity and computing power.

## D  IMAGE RECONSTRUCTION VISUALIZATION

We compare the reconstruction results on *0854* (Fig. 5), *1c55* (Fig. 6), and *0884* (Fig. 7) of our model with those of CDC (Yang & Mandt, 2023), HiFiC (Mentzer et al., 2020), ILLM (Muckley et al., 2023) and hand-crafted method VVC (JVET, 2025).

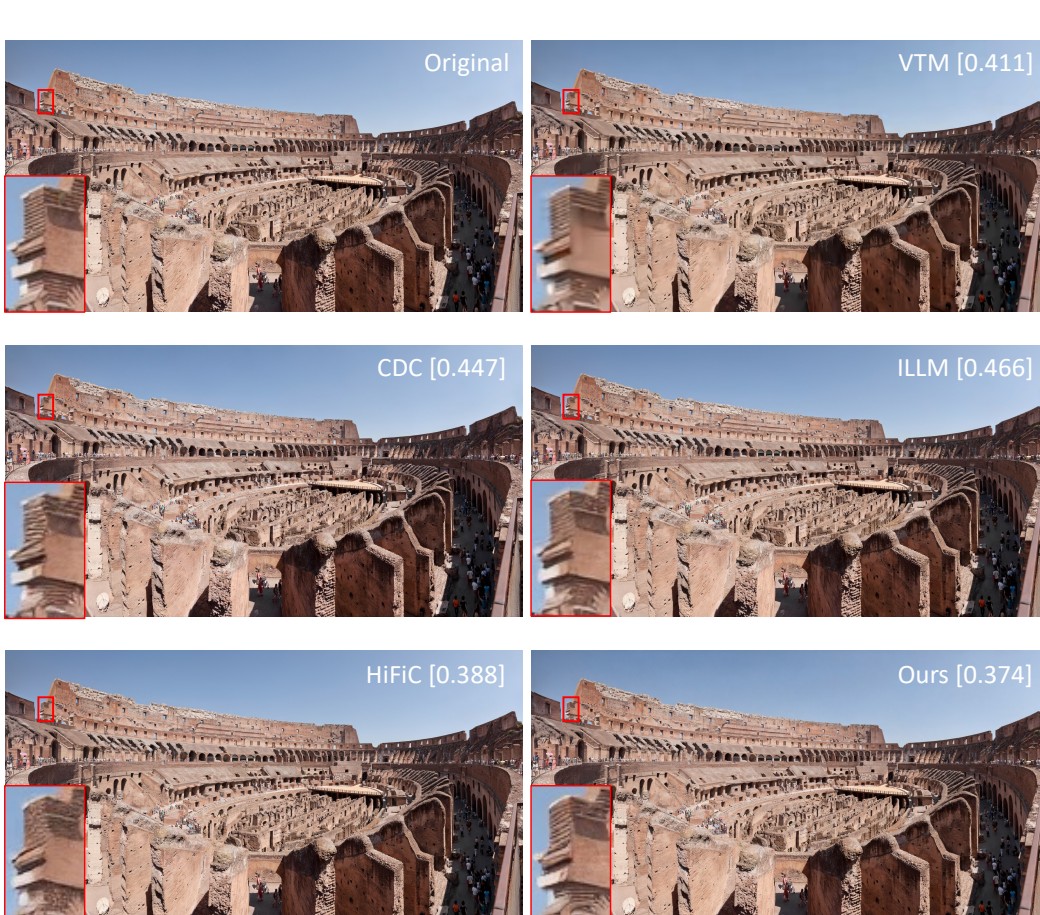

Figure 5: Visualization of the reconstructed images (*0854*) from DIV2K dataset. The titles under the sub-figures are represented as "method [bpp]".

*zoom in for better visualization*

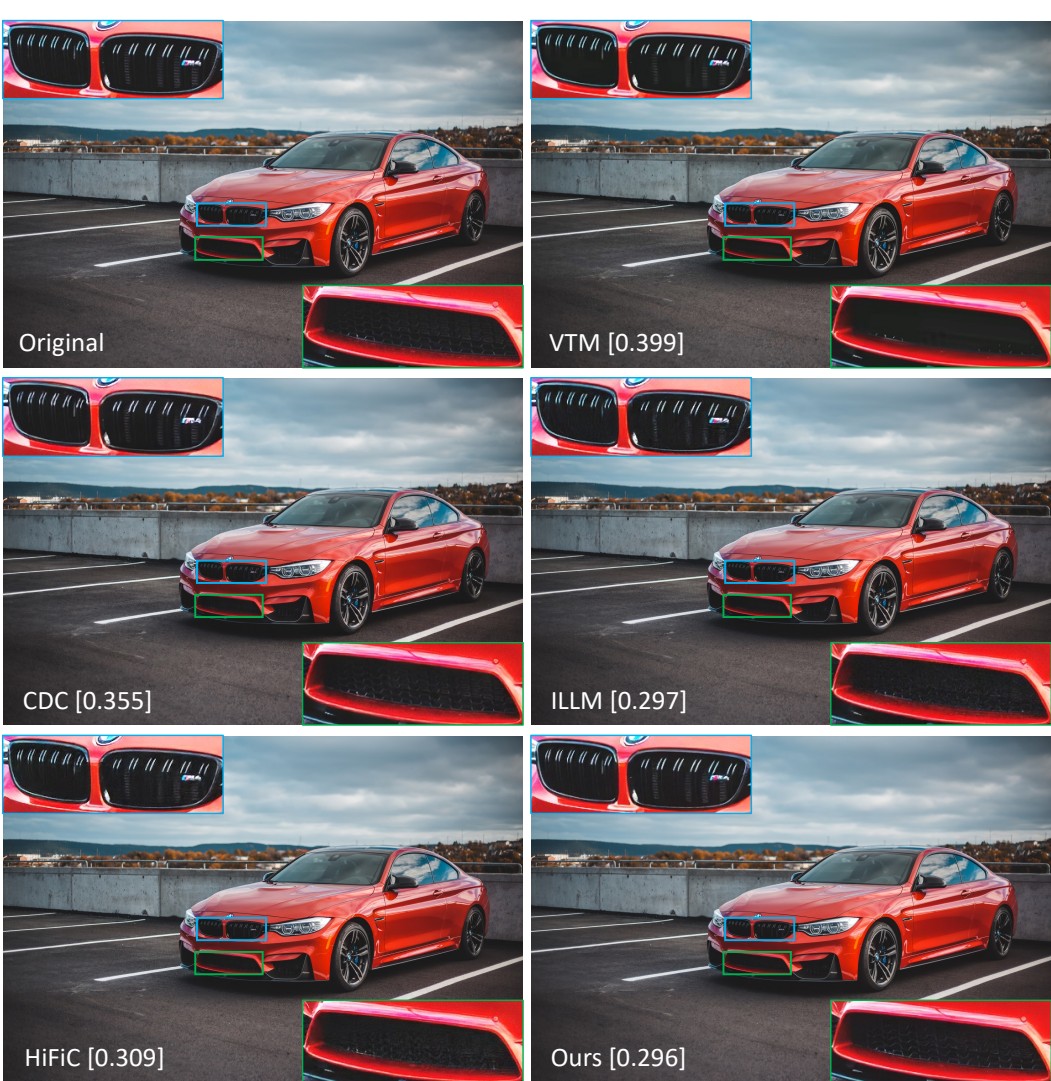

Figure 6: Visualization of the reconstructed images (*1c55*) from CLIC2020 dataset. The titles under the sub-figures are represented as "method [bpp]".

*zoom in for better visualization*

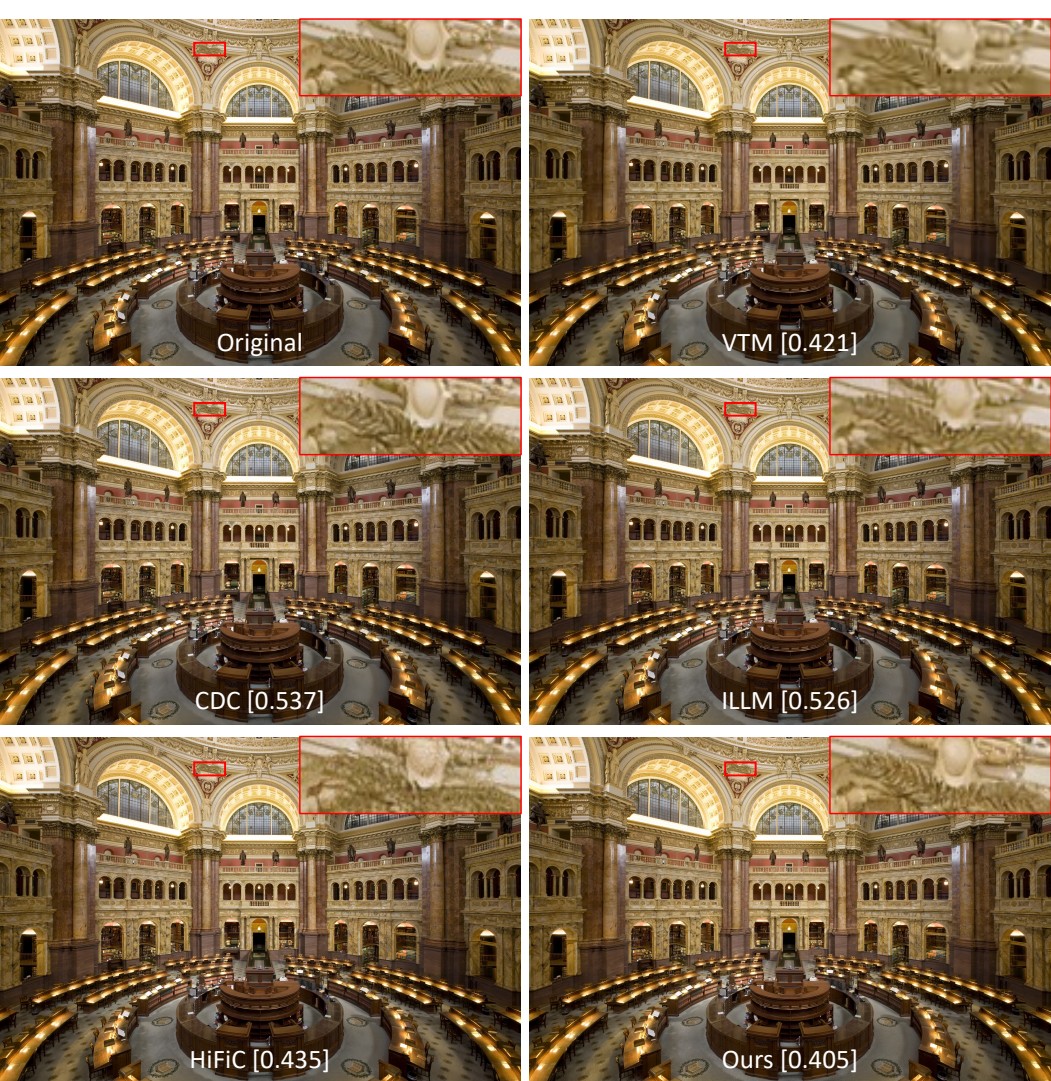

Figure 7: Visualization of the reconstructed images (*0884*) from DIV2K dataset. The titles under the sub-figures are represented as "method [bpp]".

*zoom in for better visualization*

## E  FURTHER EXPERIMENTAL RESULTS

**Additional comparisons on more datasets.**    We also compare our methods on the CLIC2020 and Kodak datasets. For the Kodak dataset, because too few images are contained, the statistical fidelity metrics (FID and KID) are invalid to evaluate the reconstructed results. The RD curves are revealed in Fig. 8 and Fig. 9.

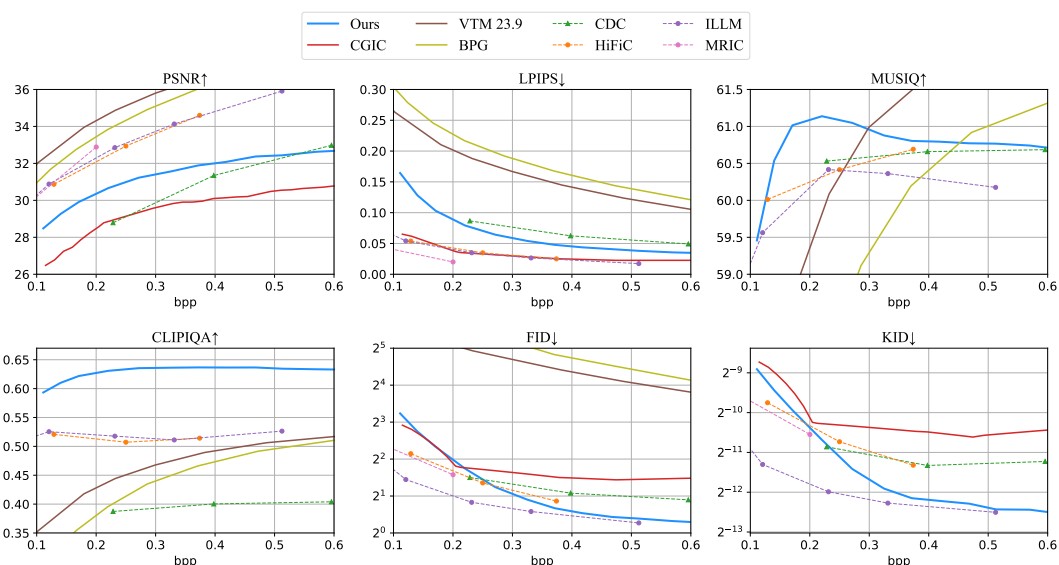

Figure 8: Comparisons of methods across various metrics on the CLIC2020 dataset.

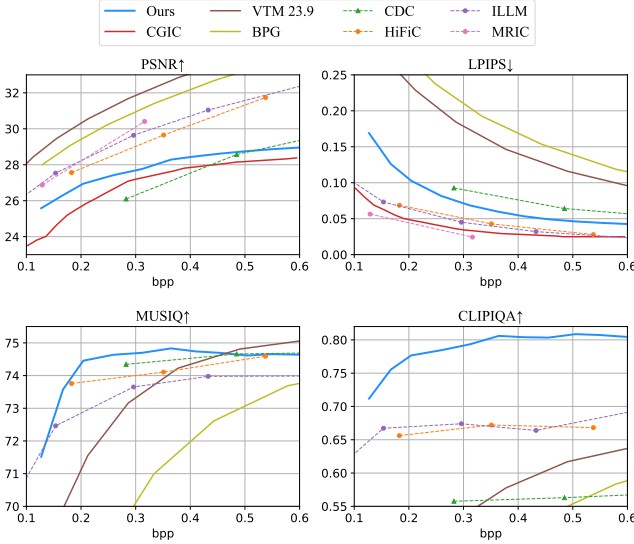

Figure 9: Comparisons of methods across various metrics on the Kodak dataset.

Table 4: Further comparisons at ultra low bitrates, evaluated on DIV2K512, the same setting as the DDCM paper.

| Method | BPP | FID | LPIPS | PSNR |
|--------|-----|-----|-------|------|
| BPG | 0.134 | 122.895 | 0.348 | 25.095 |
|  | 0.327 | 87.879 | 0.192 | 28.049 |
| CRDR-D | 0.152 | 89.766 | 0.203 | 26.618 |
|  | 0.274 | 69.466 | 0.118 | 28.762 |
| CRDR-R | 0.152 | 39.291 | 0.081 | 26.110 |
|  | 0.274 | 31.422 | 0.048 | 28.275 |
| HiFiC | 0.226 | 39.832 | 0.066 | 26.138 |
| ILLM | 0.104 | 42.591 | 0.107 | 24.579 |
|  | 0.187 | 34.103 | 0.069 | 26.206 |
| PerCo (SD) | 0.127 | 29.516 | 0.155 | 20.483 |
| DDCM | 0.149 | 28.011 | 0.132 | 22.107 |
|  | 0.309 | 26.756 | 0.114 | 22.867 |
| Ours | 0.126 | 39.709 | 0.104 | 25.739 |
|  | 0.302 | 19.191 | 0.066 | 27.418 |

**Comparison at ultra-low bitrates.** Ultra-low bitrate image compression is inherently closer to image generation tasks, where the output often does not rely on reconstructing all pixel details but instead leverages the model's generative capabilities to synthesize visually plausible images based on the limited high-level semantic information retained in the input. Such methods prioritize semantic fidelity and perceptual plausibility over the accuracy of restoring the original image. In contrast, perceptual image compression focuses on improving the perceptual quality of image compression within conventional bitrate ranges (e.g., 0.1–1.0 bpp), emphasizing visual optimization while preserving the original image's structure and details. Therefore, the two approaches differ in their objectives, difficulty, and applicable scenarios: the former aims for "foolproof" generative reconstruction, while the latter represents a perceptual enhancement of traditional compression frameworks.

However, to prove that our approach achieves a good balance on the rate-perception-distortion trade-off, we compared several ultra-low bitrate image compression methods (Careil et al., 2023; Ohayon et al., 2025) and perceptual image compression methods that support ultra-low bitrate compression (Iwai et al., 2024; Muckley et al., 2023; Mentzer et al., 2020) with our method. The table 4 shows that our method maintains the relatively high FID generation metrics in conventional perceptual compression while also demonstrating the advantage of our approach over ultra-low bitrate methods in perceptual consistency through LPIPS and PSNR distortion metrics.

# F   MORE DETAILS OF THE ENTROPY MODEL

**Modification to support multiple bit-rates.** We simply borrowed the network structure from Han et al. (2024) and modified the entropy model using the quantization scaling method described in our background section 2.2. Specifically, to support multiple bit-rates, we changed the quantization of $\hat{\boldsymbol{y}} = \lceil \boldsymbol{y} \rceil$ in the original entropy model to $\hat{\boldsymbol{y}}_q = \lceil \boldsymbol{y}/q \rceil \cdot q$ and trained using rate-variable training, i.e., sampling different $q$ values during training.

**Hyperparameter settings.** We set the channel of latent representation $\boldsymbol{y}$ as 3, and that of hyperprior $\boldsymbol{z}$ is set as 64. We adopt the evenly grouped strategy to segment the latent representation into 3 slices. The number of stacked NAF-blocks is 4, and the channel number is 256.

