# OpenReview forum: "Generative Image Compression by Estimating Gradients of the Rate-variable Feature Distribution"
_ICLR.cc/2026/Conference — ICLR 2026 Conference Withdrawn Submission_

### Official Review · Reviewer_szi9 · 2025-10-27

**Soundness:** 2
**Presentation:** 3
**Contribution:** 2
**Rating:** 4
**Confidence:** 4

**Summary:**

The paper proposes to consider compression as the forward path of a diffusion process, where rate-variable quantization is modeled as the corruption model instead of additive Gaussian noise. The proposed method uses a latent diffusion model, in the latent space of a VAE. Experiments on several datasets compare distortion measures and quality measures across several bit rates (comparison in terms of rate-perception-distortion). The abstract and Sec. 4 claim advantages across a range of metrics.

**Strengths:**

1. Recasting rate-variable quantization as a diffusion forward process is elegant and interesting.
2. 2-step decoding yields substantially lower diffusion-time than CDC and related methods; Table C/3 reports much faster decode than several diffusion baselines.
3. On DIV2K and CLIC, the method is consistently competitive/better on MUSIQ/CLIPIQA and sometimes on FID/KID.
4. The training design is clear and reproducible. Detailed hyperparameter settings and ablations are provided.
5. Results include multiple datasets, ablation studies, and comparisons with relevant baselines—reflecting solid experimental effort.
6. The diffusion-inspired compression formulation could inspire future extensions and theories that establish the diffusion paradigm as an ever better candidate for compression.

**Weaknesses:**

1. The derivation of the reverse process (section 3), to my understanding, assumes Gaussianity and applies the Tweedie-Miyasawa relation (Eq. 2) directly to uniform noise. A formal justification (or approximation argument) is missing. Also, please add a citation to equation 2 (Tweedie-Miyasawa).
2. The paper claims improvements "across a range of metrics," yet PSNR and LPIPS are often worse than baselines. Improvements are mostly in perceptual quality only. Please temper the claims and discuss the rate-perception-distortion (RPD) tradeoff explicitly. In particular, to claim that your method is better than others, it needs to improve the performance in both distortion and perceptual quality simultaneously, for a given bitrate. This is the meaning of achieving better performance: we want to get closer and closer to the rate-perception-distortion bound.
3. The discussion in the main text omits several recent diffusion-based compression methods (e.g., PerCo, DDCM), which are only compared against in the appendix. These should be cited and discussed in the main text.
4. The stochasticity and $\alpha(t)$ schedule are introduced empirically without theoretical backing.

**Questions:**

1. The authors claim that "CDC (Yang & Mandt, 2023) is the most recent state-of-the-art diffusion-based method." However, this is a little misleading. The authors compare their method to several newer diffusion-based compression methods in the appendix, such as PerCo and DDCM. Can the authors elaborate on the difference between the compared methods in the main text, and the ones in the appendix? Why are the comparisons in the main text not including all methods?
2. Can you derive (or cite) a result linking $\nabla_x \log p(x;q)$ to $(\hat{x_0} - x)q$ for uniform corruption? If approximate, how large is the induced error?
3. What is the stationary distribution of your reverse SDE when the forward process is not Gaussian?
4. Will you revise claims of "better across all metrics" to reflect perceptual superiority at the cost of distortion? This should also be reflected in the abstract.

---

### Official Review · Reviewer_8E6j · 2025-10-29

**Soundness:** 3
**Presentation:** 2
**Contribution:** 3
**Rating:** 4
**Confidence:** 4

**Summary:**

This paper presents a new framework for modeling diffusion in perceptual image compression. Using stochastic differential equations, the authors interpret the compression process as a forward diffusion process. They then train a reverse neural network to reconstruct images by reversing the compression process without using Gaussian noise initialization. The proposed method outperforms existing generative image compression approaches in many perceptual metrics.

**Strengths:**

The manuscript is well-prepared and organized. The proposed method is novel.

**Weaknesses:**

1.The paper lacks comparisons with recent generative image compression methods. Examples include TACO [1], ICISP [2], and DiffEIC [3].
[1].Lee H, Kim M, Kim J H, et al. Neural image compression with text-guided encoding for both pixel-level and perceptual fidelity[J]. arXiv preprint arXiv:2403.02944, 2024.
[2].Wei H, Zhou Y, Jia Y, et al. A Lightweight Model for Perceptual Image Compression via Implicit Priors[J]. arXiv preprint arXiv:2502.13988, 2025.
[3].Li Z, Zhou Y, Wei H, et al. Towards extreme image compression with latent feature guidance and diffusion prior[J]. IEEE Transactions on Circuits and Systems for Video Technology, 2024.
2.Figure 8 and Figure 9 should be included in the main body of the manuscript.
3.We find that the proposed method showed poor pixel fidelity, as evidenced by the PSNR metric. Maintaining pixel fidelity is crucial for generative image compression tasks. How can this be improved?
4.MS-SSIM and DIST metrics are commonly used in generative image compression methods. Please include these evaluation results.
5. In line 162, referring to Eq. 4 should be revised as referring to Eq.5.
6. The compression performance worries me so much. How to improve its performance? For example, adding the losses.

**Questions:**

See the part of weaknesses

---

### Official Review · Reviewer_4dLY · 2025-11-01

**Soundness:** 2
**Presentation:** 3
**Contribution:** 2
**Rating:** 4
**Confidence:** 4

**Summary:**

The authors start from a neat idea: what if, instead of training a diffusion model to remove Gaussian noise, we instead trained a diffusion model to reverse the process of image compression? They propose a forward process to mimic the effects of gradually more aggressive quantization of VAE latents, and a reverse process that undoes that. They demonstrate that their method achieves competitive results with common generative compression benchmarks.

**Strengths:**

The central idea of this paper is a neat one: replace the standard forward process of a DDPM with a lossy compression process, then train a network to reverse that process. I think this is an interesting and promising direction. They achieve good results against common benchmark methods for generative image compression.

**Weaknesses:**

I have two broad concerns: 1) this paper does not engage with previous work on the topic and 2) I suspect that some of its methods, although they achieve good practical results, are not mathematically sound.

1: I don't think this is the first paper to attempt to undo quantization error in VAE latents using diffusion models. I believe that [1], [2], and [3] are basically already doing this. There are probably more papers too, those are just the first 3 I encountered. I find it concerning that no such works are cited. Can you please situate your research relative to these related works, and clearly state what your novel contribution is?

2. Two details of the proposed method that I suspect may not be theoretically sound:

a) In Section 2.2, the authors assert that "The quantization operation can be regarded as
adding a uniform noise in the range of [-0.5, 0.5]". I am not so sure you can get away with this substitution? See [2], which finds that applying a diffusion model starting from discretized data yields qualitatively different results than starting from noisy data. How do the results from [2] not contradict your claim?

b) I suspect that the stochastic sampling variant of the authors' reverse process actually takes the denoising paths out of the training distribution: the training distribution comes from the forward process, and IIUC is essentually the data distribution + uniform noise. But if, during stochastic sampling, you're adding multiple independent samples of noise, that noise will start to look gaussian, which isn't what you trained on. Again, I would anticipate that this should introduce artifacts in the reconstructions, similar to those discussed in [2].

[1] [Lossy image compression with foundation diffusion models](https://arxiv.org/pdf/2404.08580)

[2] [Bridging the Gap between Gaussian Diffusion Models and Universal
Quantization for Image Compression](https://studios.disneyresearch.com/app/uploads/2025/06/Bridging-the-Gap-between-Diffusion-Models-and-Universal-Quantization-for-Image-Compression-Paper.pdf)

[3] [RDEIC: Accelerating Diffusion-Based Extreme
Image Compression with Relay Residual Diffusion](https://arxiv.org/pdf/2410.02640)

**Questions:**

Please respond my concerns from the weaknesses section. I would especially like to see some discussion of the previous works I mentioned in the background section, and how this paper relates to / differs from these approaches.

---

### Note · Authors · 2025-11-14

I have read and agree with the venue's withdrawal policy on behalf of myself and my co-authors.